# Geometric Wheat Modeling and Quantitative Plant Architecture Analysis Using Three-Dimensional Phytomers

**DOI:** 10.3390/plants12030445

**Published:** 2023-01-18

**Authors:** Wushuai Chang, Weiliang Wen, Chenxi Zheng, Xianju Lu, Bo Chen, Ruiqi Li, Xinyu Guo

**Affiliations:** 1Information Technology Research Center, Beijing Academy of Agriculture and Forestry Sciences, Beijing 100097, China; 2College of Agronomy, Hebei Agricultural University, Baoding 071001, China; 3Beijing Key Lab of Digital Plant, National Engineering Research Center for Information Technology in Agriculture, Beijing 100097, China

**Keywords:** three-dimensional modeling, three-dimensional phytomer, wheat, morphology, plant architecture

## Abstract

The characterization, analysis, and evaluation of morphology and structure are crucial in wheat research. Quantitative and fine characterization of wheat morphology and structure from a three-dimensional (3D) perspective has great theoretical significance and application value in plant architecture identification, high light efficiency breeding, and cultivation. This study proposes a geometric modeling method of wheat plants based on the 3D phytomer concept. Specifically, 3D plant architecture parameters at the organ, phytomer, single stem, and individual plant scales were extracted based on the geometric models. Furthermore, plant architecture vector (*PA*) was proposed to comprehensively evaluate wheat plant architecture, including convergence index (*C*), leaf structure index (*L*), phytomer structure index (*PHY*), and stem structure index (*S*). The proposed method could quickly and efficiently achieve 3D wheat plant modeling by assembling 3D phytomers. In addition, the extracted *PA* quantifies the plant architecture differences in multi-scales among different cultivars, thus, realizing a shift from the traditional qualitative to quantitative analysis of plant architecture. Overall, this study promotes the application of the 3D phytomer concept to multi-tiller crops, thereby providing a theoretical and technical basis for 3D plant modeling and plant architecture quantification in wheat.

## 1. Introduction

Wheat (*Triticum aestivum* L.)—based products are consumed by over 2.5 billion people globally. The demand for wheat may increase by 40% in 2023 due to continuous population growth [1]. Therefore, wheat production should be increased for sustainable food security [2]. Wheat architecture is a comprehensive expression of wheat growth and development. The configuration of wheat architecture directly affects the canopy structure of the population, which in turn affects the light distribution among leaves, ultimately influencing the efficiency of light energy use and yield of the crop [3]. Plant architecture is also crucial for selecting high-yielding lines during breeding [4].

Wheat architecture is usually characterized based on quantitative “compact or loose” descriptions. However, it is difficult to scientifically and accurately identify particular cultivar features. Most studies have described wheat architecture based on the following perspectives: the classification of wheat plants from field morphology into W, V, strip, line, and barrel types; classification based on stem-to-ground angle characteristics, such as expansive, loose, compact, and very compact; and classification based on the quantitative indicators of plant architecture, including plant height, stem and leaf angle, leaf length, leaf width, and leaf spacing. Some scholars have also proposed using convergence index of combined width and plant height [5]. Although quantitative differentiation of plant architectures can be achieved, the criteria for plant architecture classification are unclear due to the different scales of understanding these qualitative classification indicators. Furthermore, the dimensions of quantitative indicators are in 2D, which makes it difficult to accurately express and describe the spatial characteristics of wheat architecture in 3D. Architecture description toward delineation standardization, three-dimensionality, and construction precision is crucial for the development of smart agriculture, sufficient applications of plant architecture identification and description in high-throughput phenomics [6], functional–structural plant modeling (FSPM) [7], and enhanced breeding [8].

Phytomers are repetitive units of structure that function in higher plants. Phytomers, as a scale between organ and individual, are crucial for analyzing crop structure, function, and plant shape. It can also provide a better understanding of plant growth and development, and morphological construction. The concept of phytomer was first introduced by Gray [9]. To date, scholars have systematically investigated the phytomer concept and composition in winter wheat [10], rice [11], spring wheat [12], and maize [13,14]. Most present studies have mainly investigated the concept of phytomers in combination with structural and functional models of crops. For example, Dornbusch et al. [15] proposed a morphological structure-built model within the framework of the functional structure model. A 3D modeling of the geometric structure of spring wheat phytomers was achieved by introducing equation calculations. Boe and Lee [16] compared the dry matter partitioning patterns between the phytomers of two types of grass and concluded that the phytomers can partition dry matter. Most present research on the phytomer mainly describes using measurable traits, such as leaf length, leaf angle, and internode diameters, while ignoring the detailed morphometrics. Consequently, 3D phytomers were proposed for precise quantification of plant architecture and morphology [14]. Over the years, the 3D phytomer concept was extended to involve skeleton and mesh models that can present a phytomer using 3D coordinates, and can realize geometric modelling via assembling several 3D phytomers. However, maize and wheat have significantly different morphological and structural characteristics. Wheat phytomers form single-stemmed tillers, which form individual plants. Therefore, the 3D plant architecture description of wheat is more complex, making the 3D modeling method of wheat phytomers more challenging.

Geometric modeling technology of plants is crucial in crop phenomics, and is important for accurately describing the 3D information of crop morphology. It can be subdivided into three groups: (1) Rule-based methods. The 3D organ models are built using mathematical equations, and individual plant models can be derived by assembling the organs. Well-known models and software have been widely used in FSPM, such as L-studio [17], ADEL-Wheat [18], GreenLab [19], OpenAlea [20], and GroIMP [21]. Although this method can quickly build 3D models of specific plants, it is difficult to reflect the detailed morphological differences among cultivars or management strategies. (2) Parametric and skeleton-driven 3D modeling. Geometric models of plant organs are derived by morphological feature parameters characterizing the geometry combined with free curve–surface modeling technology, or using skeleton-driven mesh deformation [22]. Although this method can better restore the main morphological structure of the plant, the real morphology is not reflected in the details of the plant, such as the leaves appearing folded and twisted in different degrees, with lobes and serrations on the edges. (3) 3D reconstruction using images or 3D point clouds. Reverse engineering is used to obtain a 3D model of plants based on measurement data [23,24,25]. Although realistic plant organs can be derived using this method, the method highly relies on point cloud feature extraction algorithms, such as organ segmentation [26], skeleton extraction [27], and surface reconstruction [28]. In summary, it is evident that existing methods cannot achieve efficient 3D modeling of wheat plants with different shapes, especially for complex wheat plants in late growth stages, due to the high number of tillers and severe cross-shadowing during wheat growth and development [29,30].

In this paper, quantitative analysis of 3D wheat plant architecture was conducted using the 3D modeling method. Furthermore, this paper systematically defines the phytomer naming rules of wheat morphological stability period, proposes a 3D modeling framework of a plant based on 3D wheat phytomers, and realizes the 3D model construction of single stem and individual wheat plants through definition and digital representation. The framework was used to construct a 3D model visualization of wheat plants, thereby forming a 3D data acquisition, 3D characterization, 3D modeling of assembled wheat plants, and 3D plant architecture quantification calculation process. The technical process balances efficiency and model realism to clarify the spatial characteristics of wheat with a complex structure. Therefore, this study may provide a theoretical basis and technical support for quantifying wheat 3D plant architecture identification, accurate phenotype identification, and efficient resource utilization.

## 2. Materials and Methods

### 2.1. Experimental Site and Setup

The planting trials of 10 winter wheat cultivars were conducted in 2020–2021 at the experimental field of the Beijing Academy of Agriculture and Forestry (N 39°56′, E 116°16′). The winter wheat cultivars had significant variations in plant architecture and specific morphological characteristics parameters (Table 1). Each variety was sown in one plot (2.25 m long and 1.5 m wide) on 4 October 2020, with 0.2 m and 0.05 m spacing between rows and plants, respectively. Each variety had three replicates (total plots; 30). The wheat plants entered the filling stage sequentially in early May 2021. The wheat morphological structure reached stability. The wheat morphological structure data were obtained from 12 May to 20 May 2021.

### 2.2. Data Acquisition and Database Construction of Wheat 3D Phytomer

The whole plant was transplanted into a pot when it reached the flowering stage via field sampling. The pots were moved indoors, and water treatment was conducted during data acquisition to maintain the stability of the sample plant and the true and reductive nature of the data. The complete 3D digitized data of all leaves of the whole plant were collected using a robotic arm digitizer Microscribe i (Figure 1a). The specification of wheat phytomer data acquisition was established based on the definition of the wheat phytomer as follows: (1) Stem information data: stem thickness information was determined using three points from the root wrap-around each phytomer, whereas the upper and lower two nodes of the phytomer were recorded as inter-node length information. (2) Leaf information contained two parts: leaf surface and leaf vein skeleton information. Leaf surface information was acquired from the leaf and stem connection point, along the leaf to the sun side of each line to obtain the left edge point, leaf vein point, and right edge point (three coordinate points) to the tip of the leaf at a coordinate point for the end point, forming the leaf data points 3*n* + 1 data form (Figure 1b). Leaf vein skeleton information acquisition: several leaf vein skeleton coordinate points were obtained from the leaf and stem connection point along the leaf growth direction, ending with the leaf tip apex (Figure 1c). The distance between the data rows should not be too far, and should represent the information, such as leaf morphological changes and the highest point of the leaf. Finally, the 3D phytomer with all data points was normalized. The sections were vertically upward, and the center point of the section was reset to the coordinate origin ($H_{i}^{\mathrm{Phytomer}}$ set as 0). The phytomer azimuth $\alpha_{i}^{\mathrm{Phytomer}}$ was normalized to zero via rotational transformation, and the corresponding morphological parameters were processed accordingly. The normalized 3D phytomer template was easy to identify in plant geometry modeling.

The collection of the standardized 3D wheat phytomer data was based on the above specifications. The wheat 3D phytomer database was constructed based on the plant 3D visual resource library construction method [31]. The database contained 3D mesh model information for each phytomer, morphological parameters, agronomic parameters for each phytomer, and other information (Table 2). The 3D mesh model information was constructed using the obtained data points following the 3D digitized data acquisition specifications described above. Morphological parameters corresponding to each phytomer were extracted with each data point [32] or measured manually. The agronomic parameters were also recorded manually. The section unit morphology parameters and other information are extensible items, and additional information can be added as needed.

## 3. 3D Phytomer Naming Rules and Plant Architecture Evaluation Based on 3D Parameters

The overview of geometric modeling and quantitative plant architecture analysis methods are presented in Figure 2. First, a 3D wheat phytomer was defined and explained, followed by determination of the digital representation of different phytomers. Geometric modeling of wheat was realized by assembling 3D phytomers into a tiller or stem and into individual plants. Finally, the plant architecture vector was estimated for the corresponding geometric model of any wheat plant. A wheat plant could be modeled or reconstructed by estimating the translation and rotation matrix according to the measured data, and calling geometric templates in the 3D phytomer database. The output of the method was geometric models and their plant architecture parameters.

### 3.1. Definition of 3D Wheat Phytomer

The winter wheat phytomer in the growth and development model contains nodes and the upper space of the node: leaves, leaf sheaths, and axillary buds (Figure 3). In this paper, a 3D phytomer of wheat was composed of internodes and internodal appendages of nodes and their upper parts. The single phytomer mainly includes the internodal segment, leaf sheath, leaf blade, and panicle. The single stem of wheat contains several phytomers (Figure 1). *P_k_* represents the *k*-th phytomer, consisting of four parts: leaf blade, leaf sheath, internodal part wrapped in the leaf sheath, and joint and nodal appendages connected by the leaf sheath. Other phytomers do not have appendages if the phytomer of *k* inter-node appendages are mainly ears of wheat. The wheat phytomer is sequentially named from top to bottom based on the difference in the spatial position and physiological role of the phytomer: phytomers with spike, middle phytomers, and first phytomer (basal section). Each type of phytomer has different morphological and physiological characteristics during wheat growth and development. This is the main component factor of wheat plant formation. The basal phytomer mainly determines the growth direction of a single stem, which affects the degree of looseness of a single stemmed plant. The length of the intermediate node represents the main constituent factor of wheat plant height. The lower section of the ear represents the wheat ear and the flag leaf of wheat, and is associated with the formation of dry matter and grain yield.

The use of wheat phytomers allows the breakdown of wheat from a single plant and stem into well-structured morphological structural units. The concept of 3D phytomers of wheat can be improved using the above-mentioned characteristics of phytomers and combined with 3D data acquisition methods to realize the assembly of 3D phytomers and 3D modeling of single plants of the plant. The wheat 3D phytomers contain 2D morphological parameters and 3D spatial coordinate information. The 3D spatial coordinate information consists of a complete grid model and a simplified skeleton model.

### 3.2. Digital Representation of 3D Wheat Phytomers

The 3D phytomers of wheat were quantitatively characterized in a display manner to provide a convenient means of invocation and expression for the construction of plants and groups using the digital expression method of wheat 3D phytomers as shown below (Equation (1)):(1)$$Phytomer_{k}=\left[M_{k},S_{k},Q_{k}\right]=\{\begin{array}{l} \begin{array}{l} \left[M_{k}^{\mathrm{Leaf}},S_{k}^{\mathrm{Leaf}},Q_{k}^{\mathrm{Leaf}}\right]\\ \left[M_{k}^{\mathrm{Sheath}},\;S_{k}^{\mathrm{Sheath}},Q_{k}^{\mathrm{Sheath}}\right] \end{array}\\ \begin{array}{l} \left[M_{k}^{\mathrm{Internode}},\;S_{k}^{\mathrm{Internode}},Q_{k}^{\mathrm{Internode}}\right]\\ \left[M_{k}^{\mathrm{Appendage}},\;S_{k}^{\mathrm{Appendage}},Q_{k}^{\mathrm{Appendage}}\right] \end{array} \end{array}$$
$$Q_{k}^{\mathrm{Leaf}}=\left(H_{k}^{\mathrm{LeafBase}},\;L_{k}^{\mathrm{Leaf}},\;\theta_{k}^{\mathrm{Leaf}},\;\alpha_{k}^{\mathrm{Leaf}},\;\cdots \right)$$
$$Q_{k}^{\mathrm{Sheath}}=\left(H_{k}^{\mathrm{SheathBase}},\;L_{k}^{\mathrm{Sheath}},\;\theta_{k}^{\mathrm{Sheath}},\;D_{k}^{\mathrm{SheathMax}},\;D_{k}^{\mathrm{SheathMin}}\right)$$
$$Q_{k}^{\mathrm{Internode}}=(H_{k}^{\mathrm{InternodeBase}},L_{k}^{\mathrm{Internode}},\;\theta_{k}^{\mathrm{Internode}},\;D_{k}^{\mathrm{InternodeMax}},\;D_{k}^{\mathrm{InternodeMin}})$$
$$Q_{k}^{\mathrm{Appendage}}=\left(D_{k}^{\mathrm{NodeMax}},\;L_{k}^{\mathrm{Ear}},\;D_{k}^{\mathrm{EarMax}},\;\cdots \right)$$

$Phytomer_{k}$ denotes the *k*-th phytomer, including the 3D mesh model $M_{k}$, the 3D skeleton model $S_{k}$, and the morphological parameter set $Q_{k}$ for that phytomer.$\;M_{k}^{\mathrm{Leaf}}$,$\;M_{k}^{\mathrm{Sheath}}$,$\;M_{k}^{\mathrm{Internode}}$, and $M_{k}^{\mathrm{Appendage}}$ denote the 3D mesh models of the leaf blade, leaf sheath, internode, and appendage in the *k*-th phytomer, respectively. The detailed spatial morphological information of each component of that phytomer and each 3D mesh model contains the coordinates of all the 3D points with the phytomer and the mutual mesh connection relationships. $S_{k}^{\mathrm{Leaf}}$,$\;S_{k}^{\mathrm{Sheath}}$,$\;S_{k}^{\mathrm{Internode}}$, and $S_{k}^{\mathrm{Appendage}}$ denote the 3D skeleton models of the leaf blade, leaf sheath, internode, and appendage in the *k*-th phytomer, respectively. The brief spatial morphological information of each component of that phytomer included the leaf veins of the leaf blade, the mid-axis of the leaf sheath, and the mid-axis of the internode. Each 3D skeleton model consists of the set of 3D coordinate points of the skeleton of that phytomer. $Q_{k}^{\mathrm{Leaf}}$, $Q_{k}^{\mathrm{Sheath}}$, $Q_{k}^{\mathrm{Internode}}$, and $Q_{k}^{\mathrm{Appendage}}$ denote the set of morphological parameters of the leaf blade, leaf sheath, internode, and appendage in the *k*-th phytomer, respectively. The main morphological parameters are shown in Appendix A. Each morphological parameter set is an extensible vector. The feature parameters can be added or deleted depending on the actual demand.

### 3.3. Geometric Modeling of Single Stem and Plant Based on 3D Phytomers

Unlike maize, wheat has multiple tillers, and the mutual position relationship of each tiller has higher freedom and masking. Moreover, the phytomers of wheat are significantly different among cultivars, limiting wheat phytomer construction. Herein, a geometric modeling approach was developed for wheat phytomers, in the form of single stem (Equation (2)), and single plant (Equation (3)).
(2)$$Tiller_{ij}^{n}=\cup_{k=1}^{n}R_{ijk}\cdot T_{ijk}\cdot Phytomer_{ijk}\;$$

$Tiller_{ij}^{n}$ indicates that there are *n* single stems in the single plant with serial number *i*, with the *j*-th single stem; $Phytomer_{ijk}$ represents the *k*-th phytomer among the *n* phytomers on a single stem *j*; and $R_{ijk}\;\mathrm{and}\;T_{ijk}$ represent the rotation matrix and translation matrix corresponding to the section, respectively.
(3)$$Shoot_{i}^{m}=\cup_{j=1}^{m}R_{ij}\cdot T_{ij}\cdot Tiller_{ij}$$
where *i* indicates plant serial number. Wheat plants contain *m* single stems/tillers. $Shoot_{i}^{m}$ represents a wheat plant containing *m* unfolded leaves. $Phytomer_{ij}$ represents the *j*-th phytomer of the *i*-th plant. $R_{ij}$ and $T_{ij}$ represent the rotation and translation matrices, respectively, corresponding to the *j*-th phytomer of the *i*-th plant.

Notably, three kinds of input data were required when modelling a wheat plant. First, the tiller number *m* and phytomer number *n* in each tiller were determined. Second, 3D phytomers in the database were selected. Finally, the translation and rotation in Equations (2) and (3) were determined. For interactive modelling, these inputs are specified by users. For 3D reconstruction, the 3D phytomers are determined by similarity, and the rotation and translation matrix are calculated according to the data obtained from the target plant.

### 3.4. Spatial Parameter Extraction and Plant Architecture Evaluation Based on 3D Modeling Results

#### 3.4.1. Extraction of 3D Phenotype-Based Metrics

The 3D modeling of the plants was conducted based on the actual data of the 10 cultivars acquired via the 3D digitizer. A series of plant parameters (Table 3) were obtained from the leaf scale, 3D phytomer scale, and single stem scale based on the modeling results.

#### 3.4.2. Evaluation of Plant Architectures Based on 3D Phenotypic Indicators

The convergence index (*C*) was defined as the ratio of plant height (vertical) and spike width (horizontal) (Convergence index = plant height/spike width). The plant height and spike layer width reflect the longitudinal height of the plant and the lateral dimensions of the plant, respectively. However, the spike layer width causes errors in actual measurements due to individual subjective differences. Therefore, the spike layer projected area (*S*_area_), calculated from the reconstructed 3D model, was used instead of the spike layer width (Equation (4)). The convergence index calculation results were expanded by a factor of ten to classify the plant architecture.

Although the plant architecture convergence index can describe the differences in profiles of single wheat plants in horizontal and vertical directions, it cannot analyze quantitative differences in organ scales, such as leaf and stem types between lines. Herein, the spatial parameters of plant architecture were extracted from the spatial leaf scale, stem type scale, and single plant scale based on the modeling results. The leaf structure index (Equation (5)), the unit structure index of 3D phytomers (Equation (6)), and the stem structure index (Equation (7)) were then calculated. The plant architecture vector (*PA*) was constructed by combining the improved plant architecture convergence index (Equation (8)). The standard workflow of 3D digital data acquisition, 3D modeling, plant architecture parameter extraction, plant architecture quantitative description, and plant architecture division was formed.
(4)$$C=h*10/S_{\mathrm{area}}$$
(5)$$L=L_{\mathrm{bend}}*\theta_{l}$$
(6)$$PHY=S_{\mathrm{phy}}/N_{\mathrm{length}}$$
(7)$$S=\theta_{s\;}$$
(8)PA=(C, L, PHY, S)

## 4. Results

The detailed geometric modeling process of assembling 3D phytomers into tillers and individual plants has been described. Ten cultivars (three replicates) with different plant architecture were used to demonstrate the modeling ability and quantitative plant architecture resolution in describing various plant morphometrics.

### 4.1. Geometric Modeling Results

The 3D modeling of wheat plants was realized by combining 3D digitizer data with the 3D phytomer concept. The morphological parameters of the wheat plant were quantified using a 3D digitizer based on the 3D phytomer geometric modeling of a single wheat plant through 3D design and assembly of parts in the industrial and manufacturing fields. A template library was then formed based on the section unit. Rotation and translation matrices for individual phytomer, stem, and single plant expressions were constructed. The geometric growth points of each wheat plant originated at the tiller nodes. The number of single phytomers and single stems was determined via data collection, whereas the number of different phytomer types was determined from every single stem downward. A single stem was randomly selected and named $Tiller_{11}^{12}$ if the wheat plant with serial number 1 had eight single stems. The rule of naming wheat phytomers indicates that the phytomers of a single stem should be named from top to bottom ($Phytomer_{1,1,4}$,$\;Phytomer_{1,1,3}$, $Phytomer_{1,1,2}$,$\;\mathrm{and}\;Phytomer_{1,1,1}$) if the stem contains four phytomers. Herein, the phytomers were named based on this rule, which resulted in a template library of phytomers for the plant. Finally, the corresponding rotation matrix $R_{ij}$ was determined using the azimuth $\alpha_{i}^{Phytomer}$ of each phytomer after determining the 3D template of each phytomer of the target plant by resolving the relative position relationship among the phytomers in the 3D digitizer. The corresponding translation matrix $T_{ij}$ was also determined using the height $H_{i}^{Phytomer}$ of each phytomer to form a single stem as $Tiller_{11}^{12}$. A 3D geometric model of a single wheat plant was reconstructed after determining the rotation matrix $R_{ij}$ and translation matrix $T_{ij}$ for each single stem and specifying all parameters in Equation (2) (Figure 4).

### 4.2. 3D Modeling Results

The above 3D phytomer-based digital representation of wheat plants combined with an assembled geometric modeling process (Figure 4) was used to construct a 3D model plant for all cultivars (Figure 5). This method achieved 3D plant modeling at various resolutions. The modeling results showed that the cultivars had significant morphological differences. The combination of the plant convergence index (*C*) was further visualized for plant comparison. The stems of the cultivars converged and became compact as the plant convergence index increased (Figure 6a), while the leaves became more erect. The D1 variety had the loosest stem, with the smallest mean value of plant convergence index (*C*) and the largest degree of stem draping. Although the D10 variety had the largest convergence index, the first replication did not conform to the law of variation of plant architecture. Moreover, the D10 variety was influenced by the environment. In addition, the modeling results of the experimental replications of the D7–D10 cultivars showed the phenomenon of “re-loosening” (the part selected in the red box in Figure 5) due to the influence of the environment of the planting area, variability of the variety, and planting density of the variety. The variation of plant architecture between replications was large.

### 4.3. Evaluation of Quantitative Plant Architecture Vectors

The stem structure index (*S*) gradually decreased with increasing convergence index (*C*) (Figure 6a). Although the variation patterns of phytomer structure index (*PHY*) and leaf structure index (*L*) were similar, their variation was not significantly correlated with the convergence index (*C*). The convergence index (*C*) was about 1.95–9.56, with a maximum difference of about five times. The plant architecture of different cultivars could be distinguished because of the large difference in the convergence index. However, the source of plant architecture differences for cultivars with similar values could not be distinguished and clarified due to the small differences in the intermediate type values. Therefore, it is necessary to further refine the differences among sub-cultivars based on the other three internal structure indices. For example, the D3 convergence index (*C*) of the same intermediate variety was 4.12, whereas that of D4 was 4.29. The phytomer structure index (*PHY*) and leaf structure index (*L*) were higher in the D4 variety than in the D3 variety. Higher leaf structure index (*L*) in the D4 variety indicated that the leaf characteristics were more obvious, the combined effect of leaf pendency and stem-leaf angle was stronger, and the leaf shape was more pendulous and draped compared with that of the D3 variety, which is consistent with the 3D modeling results shown in Figure 5. In addition, the convergence index (*C*) had a large error between replications of each variety. For example, the error value of D10 cultivars reached 6.78, and the error-to-mean ratio of all cultivars was higher than the ratio of other plant architecture indices of the same variety except for the D3 variety. However, the error of other plant architecture structure indices had less variance (Appendix A) because the convergence index mainly characterized the external morphology of wheat plants, which is influenced by a combination of factors. Correlation analysis (Figure 6b) showed that the phytomer structure index (*PHY*) was significantly and positively correlated with the stem structure index (*S*) (r = 0.47). Furthermore, stem structure index (*S*) was significantly correlated with the convergence index (*C*) (r = −0.88). However, the phytomer structure index (*PHY*) and leaf structure index (*L*) were significantly and negatively correlated with convergence index (*C*), which may be attributed to the method used to calculate the composition index (*C*) and the stem structure index (*S*). The smaller the angle between the stem and the ground, the smaller the horizontal area of the plant canopy, and the larger the plant composition index (if the plant height remains unchanged).

The quantified values of plant architecture for the 10 cultivars are shown in Figure 7. The traditional plant architecture description mainly analyzes the plant architecture status empirically and qualitatively, ignoring the quantitative description, especially the 3D data. In this paper, the 3D architecture parameters extracted from the 3D modeling results were used to convert the traditional plant architecture qualitative description into quantitative descriptions based on the improved convergence index. The convergence indices (*C*) of the loose-styled cultivars, semi-compact cultivars, and compact cultivars were 1.95, 2.87–5.72, and 6.36–9.56, respectively, achieving the 3D plant architecture quantification for cultivar resolution. The plant architecture vector radar plot of each variety in Figure 7 showed the variation in architecture composition index of each variety in detail. The D1 architecture composition plot of the loose-type variety was the flattest, with the smallest convergence index (*C*) and largest stem convergence index (*S*). The plant architecture became more compact as the convergence index increased. The convergence index and the leaf structure index surrounded a larger area of the plant architecture radar plot area. The constructed plant architecture vectors were used to quantify plant architecture differences between cultivars and replications, and perform plant architecture identification and plant architecture quantification evaluation by integrating 3D information at the organ, phytomer, and individual plant scales.

## 5. Discussion

### 5.1. Development and Application of 3D Phytomer for Multi-Tiller Crops

The concept of wheat phytomers has been mainly applied in the auxiliary growth and development model and general model of cereal crops [1,10,12,33]. The 3D phytomer was first proposed in maize [14], where it was used for geometric modeling via assembling the basis morphological units. However, considering that wheat and maize have significant morphological differences, directly using the 3D phytomer on wheat is challenging, especially due to the tillers and multi-stem characteristics of wheat [14]. Furthermore, the spatial configuration of the leaves overlapping and shading each other limits the spatial description and 3D modeling of wheat plants. A previous study proposed a 3D modeling method for wheat based on leaf shape features and mesh deformation [22]. The geometric models were realized using measured data and geometric modelling, but the interactive design into diverse wheat models was not realized. In this study, this method was employed to realize an assembled 3D modeling for wheat cultivars. Notably, users can interactively adjust the rotation and translation matrix to design and derive various maize models.

This paper has the following innovations:The naming rules of 3D phytomers were clarified in a multi-tiller crop, and the assembled wheat plant modeling was completed by systematically defining the naming rules and digital expressions of 3D phytomers in wheat at the morphological stability stage and defining the digital expressions of wheat based on 3D phytomers according to the expressions (Figure 4).The 3D assembly-based modeling was realized based on phytomers, and the modeling results could observe the flag leaf characteristics, the degree of draping of other leaf layers, and the degree of single-stem aggregation among cultivars, and visually compare and analyze the sources of plant differences among cultivars (Figure 5).The plant architecture evaluation indices of each scale were calculated based on the multi-scale 3D spatial parameters of organs, phytomers, and single stems extracted from the modeling results. Moreover, the indices were significantly correlated. Finally, the plant architecture vector was constructed using multi-scale plant architecture evaluation parameters for quantitative analysis of different wheat cultivars (Figure 6 and Figure 7).

### 5.2. Advantages of Assembled Geometric Modeling Based on 3D Phytomers

Rule-based geometric modelling methods, such as L-systems [34] and automata models [35], both dual-scale and multi-scale, focus on structural variations in plant topology, ignoring the description of detailed features in crop morphology. Therefore, this method is mostly used for simulation and computational analysis among different species [36], but not for plant architecture identification to quantify plant architecture differences among cultivars. The 3D phytomer-based geometric wheat modelling method proposed here inherited the modelling efficiency and convenience of rule-based methods, and improved the morphological details. Image-based or point cloud methods obtain morphological data via multi-view images [37] or scanners. Semantic segmentation [38] is challenging, especially for complex crops [39]. The internal deficiency of modeling results for complex crops is relatively large due to the challenge of mutual occlusion between branches and leaves. The 3D phytomer-based method simplified the reconstruction process and obtains semantic phytomer models into a database. With the enrichment and improvement, the database will strongly support the geometric wheat modelling. The 3D digitizer-based method [22,40] obtains a more realistic 3D model of crops and is considered a more accurate method for describing plant structure. However, the measurement process is time-consuming and is associated with slow acquisition speed. The assembled geometric modeling method for wheat based on 3D phytomers is more convenient by comparison. Geometric models can be obtained through the 3D phytomer database when the tiller numbers, phytomer number, rotation, and translation matrix are known. The 3D results obtained using this framework could visually describe the differences in wheat plant architectures among the various cultivars and achieve highly realistic 3D modeling.

### 5.3. Assembled Geometry Modeling Based on 3D Phytomers Promotes the Study of Wheat Plant Architecture Quantification

The plant architecture classification lacks uniform indicators, and the expression is not rigorous enough [5]. Traditional measurement and 2D evaluation indices [29] separates organs from individual plants and cannot characterize the spatial relative position changes of leaves and stems of plants in 3D. In addition, the plant architecture indices are not comprehensive enough. For example, plant architecture convergence index and plant architecture height composition index [41], which are based on “stem type” for plant architecture identification, ignores the contribution of leaf type to plant architecture identification. Nevertheless, the leaves significantly influence the plant shape. Various spatial states of the leaves, combined with the stem state, can describe the full range of the plant shape. Furthermore, there are few quantitative plant indicators that do not adequately reflect plant information. For example, the plant convergence index can only define the effect of plant height and canopy width on plant shape [42]. The plant height composition index uses only stem length information, ignoring leaf information and stem and leaf angle information. Comparatively, the proposed wheat plant architecture quantification method can explicitly describe the characteristics of wheat plant morphology by considering the contribution from phytomer, stem, leaf, and individual plant. These parameters can visually compare plant architectures between cultivars, analyze the sources of plant architecture differences, and construct plant architecture evaluation indicators at various levels that can be combined with multi-scale plant architecture index for quantitative plant architecture difference analysis.

### 5.4. Shortcomings and Future Work

There are also some shortcomings in the practical application process. First, the data acquisition stage for the construction of a database of 3D phytomers of wheat is laborious. Moreover, the method focuses on the description of plant architecture, mainly on leaves and stalks, which is insufficient for describing spike morphology. Crop 3D modeling has gradually developed towards high accuracy and realism due to the increased data acquisition means and the improvement of 3D modeling methods. The proposed 3D phytomer-based 3D modeling framework for wheat can be easily combined with various real-world measurement methods [30]. A 3D phytomer database can be obtained via 3D scanning, which can be combined with commercial software or surface reconstruction and other methods to build a phytomer template library. However, further studies should assess how to obtain 3D phenotypic traits in crops for crop phenotyping research. Existing studies have shown that 3D models can accurately simulate canopy light distribution, estimate grain yield, and characterize daily radiation use efficiency (RUE) decline patterns for various canopy structures [43]. Future studies should evaluate the light interception of different wheat cultivars for 3D plant architecture groups around the 3D modeling results constructed by the proposed 3D modeling framework. This can realize the calculation of the canopy light distribution of different groups, thereby enabling screening of the single plants/groups with high light-energy utilization. Therefore, this study provides some technical support for the construction of ideal plant architecture types.

## 6. Conclusions

In this study, the 3D phytomer concept was expanded to multi-tiller plants. Taking wheat as an example crop, data acquisition standards and digital representation methods for 3D wheat phytomers were defined and explained. Geometric modelling was then realized by assembling 3D phytomers to tillers and to an individual plant. The geometric modelling method was proven to be effective in revealing the morphological differences among cultivars. On this basis, a plant architecture vector was proposed for evaluating wheat plant architecture. The vector included four indicators associated with the phytomer, leaf, stem, and whole plant. Compared to traditional qualitative plant architecture evaluation methods, the proposed vector presented detailed plant architecture differences in different scales. Overall, this approach is expected to provide a theoretical basis and technical support for the quantitative study of multi-tiller plants.

## Figures and Tables

**Figure 1 plants-12-00445-f001:**
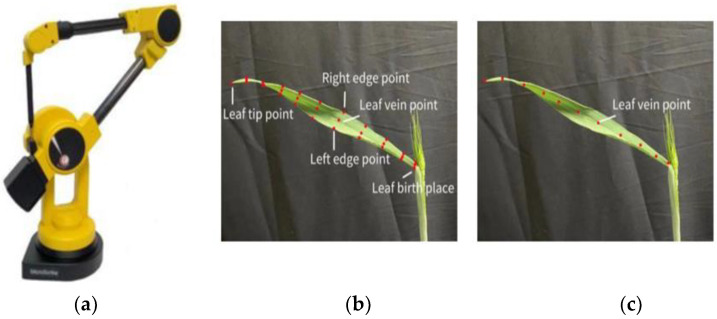
Data acquisition instrument and data acquisition schematic [22]. (**a**) The 3D digital data acquisition instrument; (**b**) leaf surface; (**c**) leaf vein skeleton.

**Figure 2 plants-12-00445-f002:**
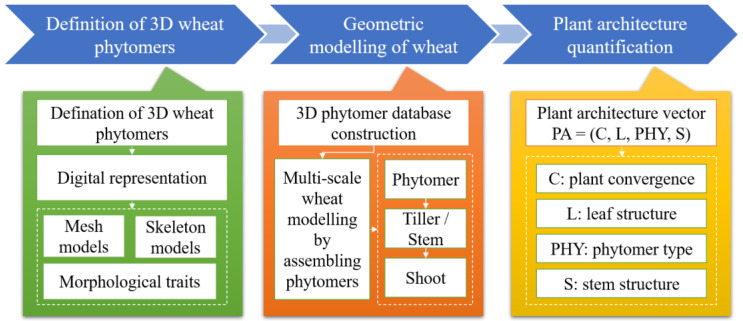
Overview of 3D phytomer-based geometric modeling and plant architecture quantification of wheat.

**Figure 3 plants-12-00445-f003:**
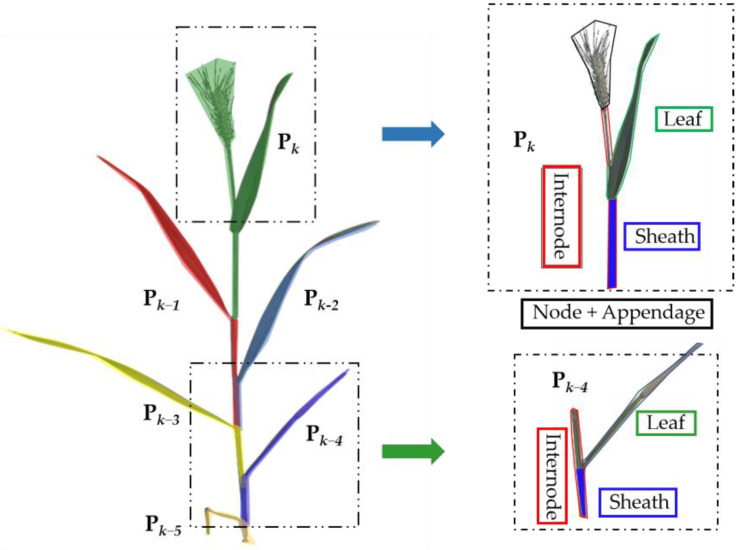
Schematic of the definition of wheat phytomers. Wheat phytomers include leaf blade (green box), leaf sheath (blue box), and node attached to the leaf sheath and appendages on the node (red box). $P_{k}$ represents the phytomer with wheat ears; *P**_k_*_−*4*_ represents the interdigitated phytomer.

**Figure 4 plants-12-00445-f004:**
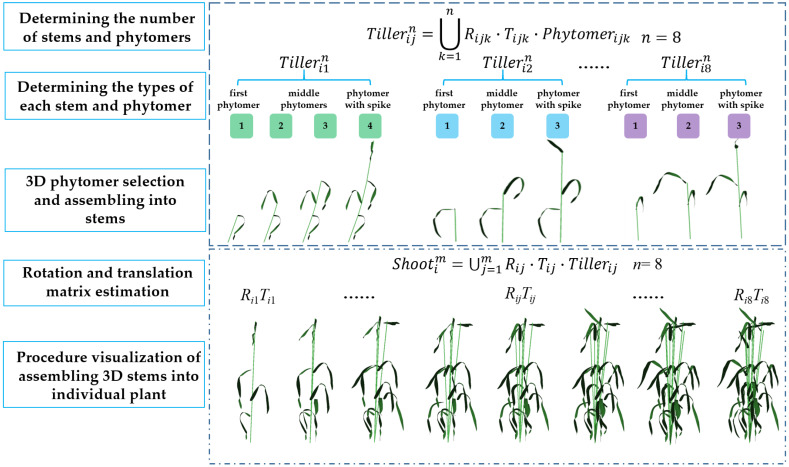
Geometric modeling of assembled wheat plants: from 3D phytomers to tillers and to an individual plant. In the figure above, the wheat plant consists of eight tillers, with each tiller having several phytomers. A tiller was assembled by selecting 3D phytomer templates, and determining rotation and translation matrices. Similarly, a wheat shoot was assembled by determining the rotation and translation matrices of the already constructed tiller models.

**Figure 5 plants-12-00445-f005:**
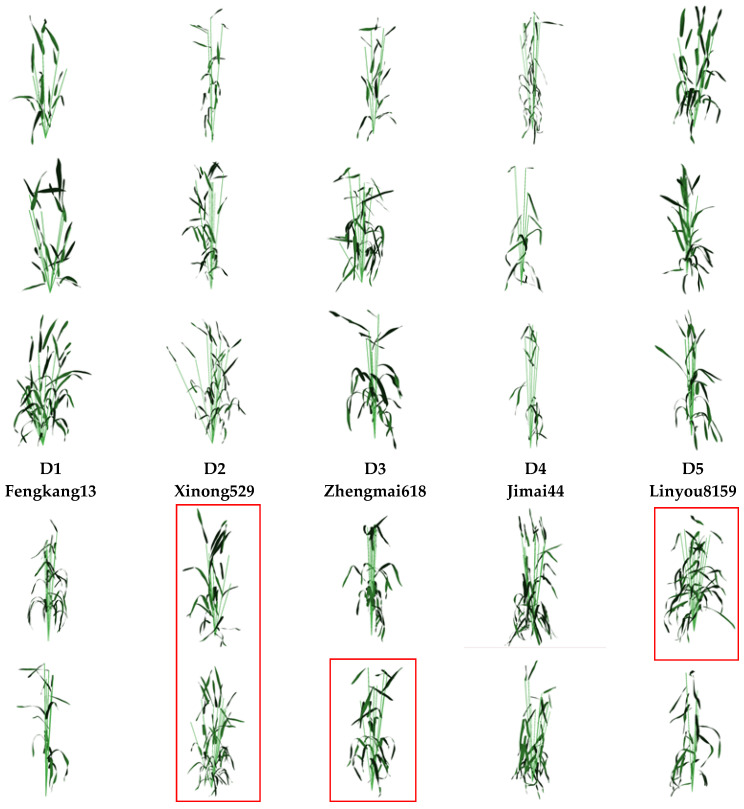
Modeling results of 10 wheat cultivars. The red boxes indicate wheat plants that are not coincident to plant architecture variation.

**Figure 6 plants-12-00445-f006:**
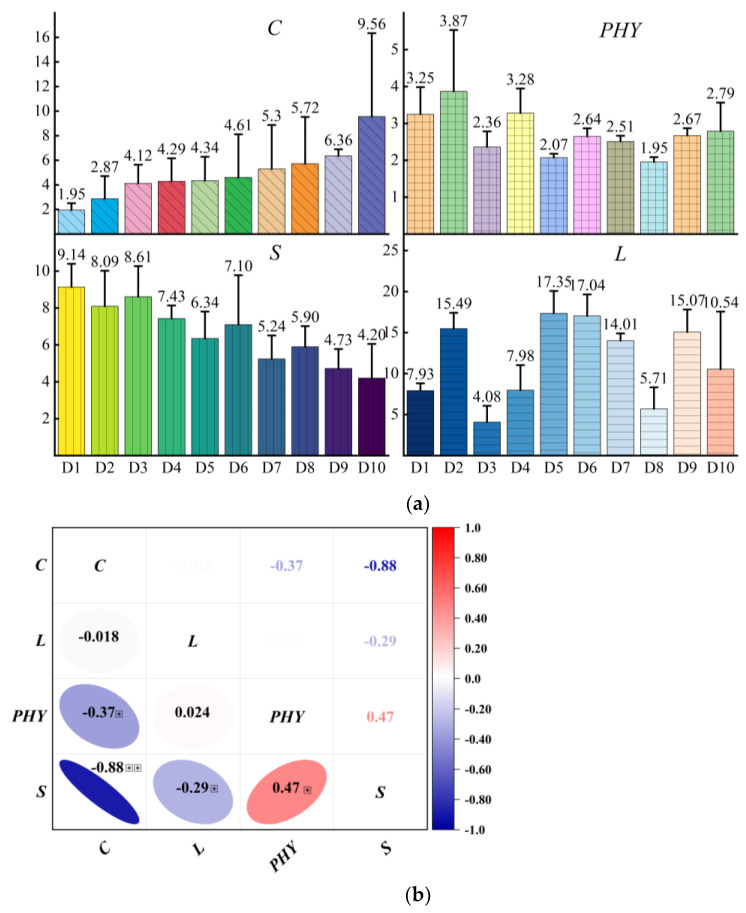
(**a**) The trend of plant architecture vectors consisting of four plant architecture indices (*C, PHY, S,* and *L*). (**b**) Correlation among plant architecture vectors.

**Figure 7 plants-12-00445-f007:**
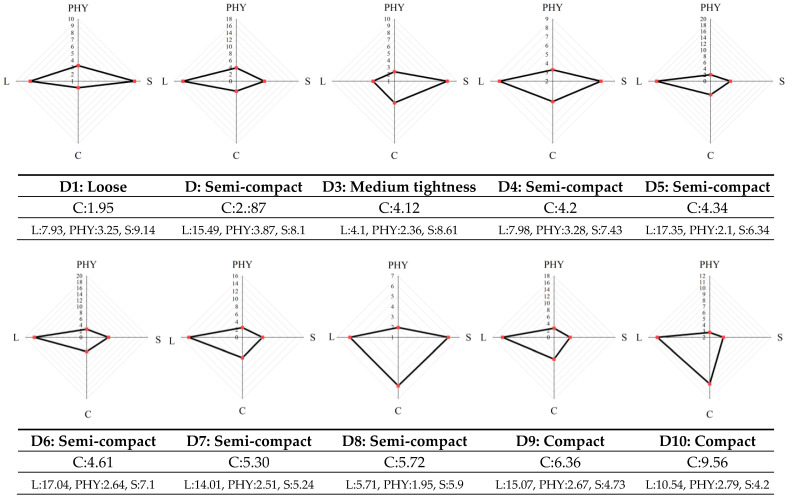
Quantitative plant architecture analysis.

**Table 1 plants-12-00445-t001:** The cultivars used in the test and their plant architecture characteristics.

Ser Number	Name	Qualitative Plant Architecture Characteristics	Plant Height, Flag Leaf/Leaf Characteristics
D1	Fengkang13	Loose	100 cm; leaves upraised
D2	Xinong529	Semi-compact	80.4 cm; flag leaf broad and long, spreading
D3	Zhengmai618	Medium tightness	76.4 cm; flag leaf broad and long, upstroke
D4	Jimai44	Semi-compact	89.4 cm; flag leaf upstroke, short and wide
D5	Linyou8159	Semi-compact	80 cm;—
D6	Xinong979	Semi-compact	75 cm; leaves upwardly inclined
D7	Jinuo116	Semi-compact	78.9 cm; flag leaf upraised
D8	Jimai38	Semi-compact	74 cm;—
D9	Jimai17	Compact	75 cm; leaves upstroke
D10	Xingmai23	Compact	72.6 cm;—

**Table 2 plants-12-00445-t002:** 3D phytomer database keywords.

Type	Keywords
phytomer agronomic parameters	phytomer type, variety, growth period, ecological point, density, row spacing, water and fertilizer treatment, stem order
phytomer 3D model information	stores paths, 3D model names, number of vertices, number of meshes
phytomer morphological parameters	spike length, spike direction vector flag leaf, middle leaf, first leaf: leaf length, leaf width, stem, and leaf angle phytomers with spike, middle phytomers, first phytomer: node length, node direction vector stem vector projection angle, stem vertical direction angle number of tillers per plant, number of nodes per stem
other information	number, access time, access person, entry person

**Table 3 plants-12-00445-t003:** Multi-scale phenotypic parameters used in plant architecture quantification.

Parameter Scale	Parameter Name	Identifiers	Description	Calculation Method
organ	Blade curvature	*L* _bend_	Quantify the overall curvature of the blade	Leaf base to leaf tip length/leaf length
	Stem and leaf angle	*θ* _l_	Quantify the degree of leaf uprightness	The angle of the stem vector and the leaf vector
3D phytomer	Phytomer envelope area	*S* _phy_	Quantification of the characteristics of the blade in the vertical direction with respect to the internodes	If there is no spike, the internode length × distance from the leaf tip point to the internode × maximum leaf width. If there is a spike, the total length of the internode and spike × distance from leaf tip point to internode × maximum leaf width
	Average internode length	*N* _length_	Quantifying the characteristics of leaves and internodes in space occupation	Distance between adjacent nodes
Single-stem	Single stem and vertical normal angle	*θ* _s_	Quantifying the degree of draping of single plants	Single stem and vertical vector angle
	Plant height	*h*	Plant height	Vertical height from the ground to the top of the spike
	Spike layer area	*S* _area_	Single plant spike layer area	Average spike layer area

## Data Availability

Not applicable.

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
