# Peer review of "Geometric Wheat Modeling and Quantitative Plant Architecture Analysis Using Three-Dimensional Phytomers"

_plants, 2023, doi:10.3390/plants12030445_

Round 1

Reviewer 1 Report

Dear Authors,

the manuscript is interesting and should certainly be published, but not in the current form. First of all, Introduction is far too general, there are no specific examples of methods used and results obtained by other researchers.

Methods are presented too generally, it is not known what the input data are, how they were tested, what methods were used to assess the accuracy and verify the models used.
The Discussion of the results must contain about 40% of all citations to emphasize the comparability of the results obtained with the achievements of other researchers.

The current version of the manuscript contains too few references, not only in the Discussions, but also in the Introduction there should be definitely more citations.

Authors must add Conclusions.

The current version of the manuscript looks like a scientific report on the conducted experiment, there are definitely no references to the literature and justification of the selected methods.

Many more detailed comments are contained in the attached manuscript.

With best regards

Reviwer

Author Response

Dear reviewer,

We very much appreciate your careful reading of our manuscript and valuable suggestions. We have carefully considered the comments and have revised the manuscript accordingly. The modified contents can be identified in red fonts.

** We have reorganized the introduction. Many well-known statements were deleted and related references have been supplemented. 12 new references were added in the introduction and discussion section to give more details about methods and achievements as suggested. However, to the best of our knowledge, there are little achievements of other researchers to solve out the problem, especially about 3D phytomer-based plant modelling and plant architecture quantification.

** The innovative methods were presented in section 3. We supplemented an overview and flowchart in the beginning of this section to make it clearer than the previous version. “A wheat plant could be modeled or reconstructed by estimating the translation and rotation matrix according to the measured data, and calling geometric templates in the 3D phytomer database. The output of the method was geometric models and their plant architecture parameters.” Besides, we didn’t use verify models. There are two reasons here. First, the visualization results were the best way to demonstrate the modelling results. Second, there is none plant architecture quantification methods to validate the quantitative plant architecture vectors, and the values in the vectors are calculated, which are unable to measure.

** We modified the discussion section to enhance our innovation. Related references were supplemented and 40% of all the citations have been reached.

** We have added the conclusions section.

** We have revised the abstract to make it shorter, and the methods and achieved results have been enhanced.

** All formulas were cited in the main text.

** We have supplemented a short introduction in the beginning of the results section.

** More descriptions were supplemented in the legends of several figures.

** Figure 6 have been reorganized to make it clear.

** We have updated all the references as suggested.

Reviewer 2 Report

The article is devoted to a very topical issue of obtaining 3D data on the architecture of wheat plants. The authors propose a very original technique. The manuscript is very well written and structured and I was a pleasure to read it. I have only a few suggestions for improving the presentation of the results in the figures.

Figure 2 It is worth showing the leaf blade (green box) and leaf sheath (blue box) more clearly. Maybe you should make the outline thicker.

Figure 4 It seems to me that it is better to sign the names of the varieties in this figure. And in the caption indicate what the red frame means.

Author Response

Dear reviewer,

We very much appreciate your careful reading of our manuscript and valuable suggestions. We have carefully considered the comments and have revised the manuscript accordingly. The modified contents can be identified in red fonts.

** We made the outline thicker as suggested in figure 3 (the original figure 2).

** We have supplemented the cultivar names in the figure, and explained the reed box frame in the legend of the figure.

Round 2

Reviewer 1 Report

Dear Authors,

thanks for the revision of the manuscript, which looks much better, but there is still a deep need to add much more detail, e.g., to the Introduction (need to add more detailsoriented on used methods and achieved results by other researchers, who solve out similar research problems, because readers need to know more details about current solutions). Your input data are described too generally. The same in case of the Discussion (no direct comparisons between your achievements and references). The Conclusions are too general.

Kind regards

Reviewer

Author Response

Dear reviewer,

We very much appreciate your careful reading of our manuscript and valuable suggestions again. The suggestions really help us to improve our manuscript. We have carefully considered the comments and have revised the manuscript accordingly. The modified contents can be identified in red fonts.

We revised the introduction section, especially the subsection about plant phytomer and geometric modelling methods. We deleted the general introductions and added more details oriented on the used methods as you suggested.

The input and output data were supplemented at the end of section 3.3.

In the discussion section, we also deleted some general descriptions that were not useful for comparison, and supplemented direct comparisons between the proposed method and related references.

We rewrote the conclusion section and hope the current version is well.